# Multimodal Unlearnable Examples: Protecting Data against Multimodal Contrastive Learning

## ABSTRACT

Multimodal contrastive learning (MCL) has shown remarkable advances in zero-shot classification by learning from millions of image-caption pairs crawled from the Internet. However, this reliance poses privacy risks, as hackers may unauthorizedly exploit image-text data for model training, potentially including personal and privacy-sensitive information. Recent works propose generating unlearnable examples by adding imperceptible perturbations to training images to build shortcuts for protection. However, they are designed for unimodal classification, which remains largely unexplored in MCL. We first explore this context by evaluating the performance of existing methods on image-caption pairs, and they fail to effectively build shortcuts due to the lack of labels and the dispersion of pairs in MCL. In this paper, we propose Multi-step Error Minimization (MEM), a novel optimization process for generating multimodal unlearnable examples. It extends the Error-Minimization (EM) framework to optimize both image noise and an additional text trigger, thereby enlarging the optimized space and effectively misleading the model to learn the shortcut between the noise features and the text trigger. Specifically, we adopt projected gradient descent to solve the noise minimization problem and use HotFlip to approximate the gradient and replace words to find the optimal text trigger. Extensive experiments demonstrate the effectiveness of MEM, with post-protection retrieval results nearly half of random guessing, and its high transferability across different models. Our code is available on the anonymous website.

## CCS CONCEPTS

• **Security and privacy** → **Privacy protections**; • **Computing methodologies** → **Computer vision**; **Machine learning**.

## KEYWORDS

Data Protection, Privacy Protection, Unlearnbale Examples, Poisoning Attack, Backdoor Attack, Multimodal Contrastive Learning

## 1 INTRODUCTION

In recent years, there has been a growing interest in multimodal models among researchers in the community [2]. Traditional methods have primarily focused on analyzing a single modal of data. However, with the rise of multimodal learning, different types of data, such as text, images, and audio, are being combined into a

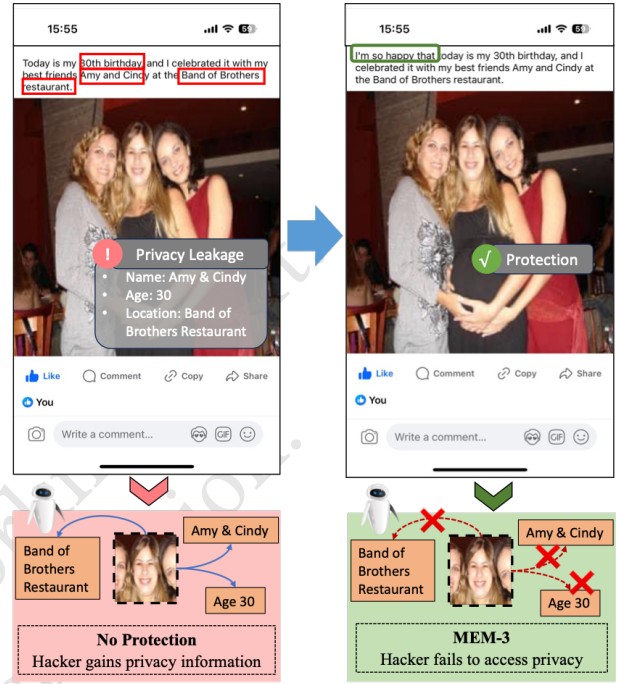

**Figure 1: Posts on Facebook inadvertently leak personal information. Utilizing MEM-3 to protect data can prevent unauthorized models from accessing private features.**

unified framework. One of the most popular approaches for multimodal learning is Multimodal Contrastive Learning (MCL), as demonstrated by models such as CLIP [31] and ALIGN [21]. These models are trained with a contrastive loss, which encourages the correlation between pairs of images and captions while also keeping them distinct from unrelated pairs. This approach reduces the need for extensive manual annotation of training data and allows the use of larger datasets that contain millions of examples. MCL has shown promise in various applications, including image classification [31], image captioning [24, 28], image generation [25, 30].

Training of high-performance multimodal models is highly dependent on large amounts of multimodal data, often sourced from publicly available datasets such as CC12M [4], YFCC100M [39], and LAION5B [34]. However, as the demand for larger datasets continues to surge in the future, these datasets may be still insufficient. Consequently, malicious actors may resort to unauthorized data acquisition from the web or engage in the crawling of user posts on social networks for commercial training purposes. However, these datasets often contain significant amounts of sensitive personal information, raising concerns among people about the potential unauthorized use of personal data and the leakage of user privacy information.

A series of recent works make efforts to prevent unauthorized usage in image classification by making the image unexploitable. Specifically, they poison the data with some imperceptible perturbations, creating 'shortcuts' [45] in the training process that hinder the models from learning the features of the images [46]. This kind of attack is called *availability attacks* or *indiscriminate poisoning attacks*, and these poisoned training data are called *unlearnable examples*. These unlearning methods can be broadly classified into two categories: model-free and model-based attacks. Model-free attacks usually generate unlearnable noise at the pixel level without any knowledge of clean data and directly create shortcuts between image noise and labels, such as LSP [45] and CUDA [32]. Due to their direct association with labels, these patterns often exhibit high efficiency. Model-based attacks typically generate noise through surrogate models. The surrogate model learns the features through the training phase and generates the feature-level noise, such as Error-Minimizing [20] and Adversarial Poisoning [13]. However, there has yet to be research to consider protecting multimodal data in the context of MCL.

We are the first to consider a scenario focused on generating multi-modal unlearnable examples against privacy risks associated with MCL. In this context, we concentrate on image-text pairs as a representative multimodal dataset. Users are assumed to frequently share personal photos with text on social media platforms like Facebook, including some private identity information such as faces, names, phone numbers, and addresses. Currently, hackers attempt to collect large amounts of such image-text pairs from the Internet and utilize MCL techniques to train modern foundational models, as illustrated in the left segment of Fig. 1. These models inadvertently capture user's private information and facial characteristics, leading to potential privacy leakage. Protectors aim to prevent the unauthorized exploitation of these sensitive data by performing unlearning methods on multimodal data. These methods aim to render models trained on such multimodal unlearnable examples incapable of accessing users' privacy features, while not impeding users' social interactions after posting images and text, as depicted in the right segment of Fig. 1.

An intuitive idea to generate unlearnable multimodal examples involves extending the unlearning methods for image classification to MCL. However, we explore the performance of these methods on multimodal data, and all of them fail to present effective protection due to increased data modalities or dispersion of data pairs. For model-free attacks, they fail to generate specific noise patterns that strongly correlate with the category for a shortcut due to the lack of certain labels in the image-text pair. For model-based attacks, while it may be feasible to optimize noise to build shortcuts with clean captions, their efficacy is significantly diminished. Our analysis was primarily attributed to the dispersion of the data pairs, which presents challenges in learning the noise pair and captions compared to the images and labels in classification, as depicted in Fig. 2. Therefore, establishing more efficient shortcuts is key to generating effective multi-modal unlearnable examples.

In this paper, we propose a novel optimization framework that efficiently generates unlearnable multimodal examples for image-caption pairs. We first adopt the Error-minimizing (EM) [20] framework as a basis for our attack, and we consider optimizing the noise along with an additional text trigger. Specifically, following adversarial triggers in NLP [41], we propose to add short text sequences as triggers in the front of the clean caption, which does not affect the understanding of the text by the user. Therefore, it can be formulated a multi-step minimization problem to optimize the noise-text trigger pair and build the shortcut between them, which we dubbed Multi-step Minimization (MEM), to prevent the unauthorized model from learning the features of images and captions. During optimization, we consider adopting projected gradient descent (PGD) [27] to solve the noise minimization problem and use the HotFlip [9] method to approximate the gradient and replacement strategy in [41] to select the optimal trigger in the text minimization problem. Through extensive experiments, we verify that unlearnable examples generated by MEM provide better protection and exhibit transferability across different hacker models.

In summary, our main contributions are:

- We are the first to consider a new scenario called multimodal data protection, which aims to prevent multimodal personal data on social media from unauthorized MCL, and we take the image-text pair as examples.
- We analyze the limitations of previous methods extended to multimodal contrastive learning, attributed to the increase in modality and the dispersion of the caption features.
- We propose a Mutli-step Error Minimization (MEM) to generate effective multimodal unlearnable examples, which leverages an additional optimized text trigger for better convergence at the basis of Error-minimizing (EM).
- Extensive experiments are conducted to verify the effectiveness of our method with different datasets. In addition, we present a practical case study on face privacy protection within a fine-tuning scenario.

## 2 RELATED WORK

### 2.1 Multimodal Contrastive Learning

Initially designed for self-supervised representation learning in unimodal contexts, contrastive learning methods aim to improve the agreement between views augmented differently in the same instance while reducing the agreement between views of distinct instances [5, 8, 16, 18, 29]. Recently, these techniques have been extended to multimodal domains, notably in the context of paired image-text datasets. Multimodal contrastive models like CLIP [31] and ALIGN [21] have undergone extensive pretraining on vast datasets including hundreds of millions to billions of image-text pairs. Their objective is to maximize the agreement between representations of matched image-caption pairs while minimizing agreement for non-matched pairs. As a result, these models have exhibited exceptional performance in zero-shot classification tasks and have shown robustness to distributional shifts.

### 2.2 Poisoning Attacks

Data poisoning attacks aim to disrupt the model training process by processing the training dataset, resulting in a significant increase in test errors for some specific samples during the testing phase. [7, 36]. A common type of data poisoning attack is the backdoor attack [7, 15, 26], which typically involves injecting triggers into training samples, leading to misclassification of images containing these

trigger patterns during testing. However, it typically only affects samples with trigger, while clean samples remain unaffected and can be correctly classified. Recent work also explores poisoning attacks on multimodal contrastive learning (MCL). Yang et al. [43] study the poisoning attacks against multimodal models in both visual and linguistic modalities. Carlini and Terzis [3] introduced a framework that effectively poisoned CLIP models with backdoor attacks. In addition, some works aim to train a robust CLIP model against data poisoning and backdoor attacks [1, 42]. Due to the time-consuming of training a high-performance model with a large-scale dataset, we will follow the experimental setting of these works to train a rather simpler model with a small dataset in this paper.

### 2.3 Unlearnable Methods

Unlearnable methods aim to protect data from unauthorized training of the classification model by introducing imperceptible noise to the images, which is also a special type of poisoning attack. Models trained on such unlearnable examples typically exhibit a notable decrease in accuracy on clean test sets. This attack is also known as the availability attack or indiscriminate poisoning attack.

There are two categories to generate unlearnable examples in image classification. The first category involves model-based attacks, which require a surrogate model to guide the perturbation generation [12, 37]. The error minimization (EM) [20] minimizes the classification error of images on a surrogate classifier and iteratively updates the surrogate model with these perturbed images. Fowl et al. [13] propose adversarial poisoning, which contains targeted adversarial perturbation (TAP) and untargeted adversarial perturbation (UAP), which uses adversarial examples as poisoned data to make the model unlearn the features. TAP presents an effective protection for the image dataset. While model-based methods are potent, they are often computationally demanding. Some findings reveal that these approaches can be easily neutralized by adversarial training (AT). To address this, several works attempt to leverage robust protection against AT [11, 14]. Model-free methods do not require a surrogate model to generate noise for images. Yu et al. [45] empirically investigate various unlearnable methods and show that all of them use these spurious features to create a shortcut in the model. They also propose the Linear-sperate Synthetic Perturbation (LSP) in response to this characteristic and show great effectiveness. Autoregressive poisoning [33] proposes a generic perturbation generation with each class that can be applied to different datasets, which is a series of dataset-independent perturbations. CUDA [32] is generated using controlled class-wise convolutions with filters that are randomly generated via a private key. These previous works focus on image classification using cross-entropy loss, and He et al. [17] explore the unlearnable examples for unsupervised contrastive learning and discover that extended EM and TAP methods can still effectively protect against unsupervised learning. However, when extending them to multimodal contrastive learning, their effectiveness diminishes due to the increased modality.

## 3 MULTIMODAL UNLEARNABLE EXAMPLES

### 3.1 Preliminaries

Our scenario involves two main parties: the *protector*, and the *hacker*. We assume that the protector is aware that multimodal privacy data may be unauthorized used by hackers. Consequently, the protector takes measures to render the samples unlearnable by performing operations on the data set. Subsequently, the protector releases these unlearnable multimodal examples to the Internet. Then, when hackers crawl the multimodal data from the web to train multimodal models from scratch with an initial model or fine-tune the pre-trained model, these unlearnable examples will prevent the model from learning the features of private data, and present a poor representation of features across modalities. In this paper, we take the image-text pair as multimodal data as an example, and choose the CLIP [31] as the model used by hackers, which is the most representative multimodal contrastive learning framework.

Here, we formulate our problem: we begin by considering a personal image-caption dataset $\mathcal{D} \subset \mathcal{I} \times \mathcal{T}$ that comprises pairs $(I_i, T_i)$, where $I_i$ is an image and $T_i$ is the associated caption. The protector aims to generate an unlearnable set $\mathcal{D}_u$ that contains the unlearnable image-text pair $(I_i', T_i')$, and make a CLIP model $f^*$ trained on it generalize poorly on the clean distribution $\mathcal{D}$:

$$\arg\max \mathbb{E}_{(I,T)\sim\mathcal{D}} \left[ \mathcal{L}\left(f^*(I,T)\right) \right],$$
$$\text{s.t. } f^* \in \arg\min_f \sum_{(I',T')\in\mathcal{D}_u} \left[ \mathcal{L}\left(f(I',T')\right) \right] \quad (1)$$

### 3.2 Limitation of Existing Works

Directly solving Eq. 1 is intractable for deep neural networks, and recent works have designed multiple approximate solutions. We can follow these works and extend them to MCL. However, all model-free methods for classification fail to generate image noise here because these methods aim to find a series of specific noise patterns for images related to one certain class, while there is no label in the image-caption pair data. Therefore, only model-based methods can be applied to the MCL, and we extend two typical methods to generate unlearnable multimodal examples. Specifically, they leveraged the addition of a set of perturbations $\Delta$ on the images and employ an $l_\infty$ to bound for each noise $\delta$.

*3.2.1 The Error-minimizing Noise. (EM).* Huang et al. [20] propose a bi-level objective to generate perturbations $\delta$ on the images with a surrogate model, which aims to minimize the loss of the surrogate model on training data. We denote image noise by $\delta$ and the surrogate model by $f$. Therefore, we can apply it to the multimodal unlearnable examples with the CLIP loss $\mathcal{L}$, and the objective is the following:

$$\arg\min_{\delta\in\Delta} \mathbb{E}_{(I,T)\sim\mathcal{D}} \left[ \min_f \mathcal{L}\left(f(I+\delta,T)\right) \right]. \quad (2)$$

*3.2.2 Untargeted Adversarial Perturbation. (UAP).* Instead of employing bi-level objectives, Fowl et al. [13] demonstrate that the common objectives used for generating adversarial examples are sufficient as unlearnable perturbations. They utilized untargeted adversarial perturbations (UAP) and targeted adversarial perturbations (TAP) as examples of unlearnable perturbations. However, extending TAP to image-caption data is challenging due to the difficulty in selecting the desired target goal for the caption. Therefore, we can only construct UAP $\delta$ using the following objective:

$$\arg\max_{\delta\in\Delta} \mathbb{E}_{(I,T)\sim\mathcal{D}} \left[ \mathcal{L}\left(f^*(I+\delta,T)\right) \right]. \quad (3)$$

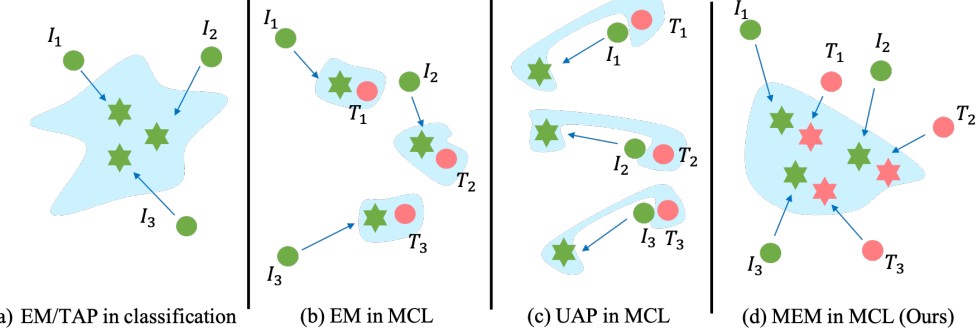

**Figure 2: Comparison of different methods in classification and multimodal contrastive learning (MCL).** $I_i$ denotes the image, and $T_i$ is the paired caption. The blue area is the expected decision boundary of the models trained on unlearnable examples.

However, from Table 1, we experimentally found that although EM and UAP can be applied to image-caption pairs, they fail to achieve highly effective protection, especially UAP. We explore the reasons for the decline in the effectiveness of these methods from image classification to multimodal contrastive learning. In image classification, EM and TAP optimize the images with the same label to converge in a feature space, resulting in the model easily capturing these additive noises and learning the correlation with labels, as shown in Fig.2(a). However, in multimodal contrastive learning (MCL), to effectively apply the EM and UAP methods, the direction of the optimized image noise must relate to the features of the caption, causing the image features to become either close to or far away from these features. Nevertheless, the caption features of different pairs may be widely dispersed in the image-caption dataset. Consequently, as illustrated in Fig. 2 (b) and (c), it becomes more challenging for the model to capture the correlation between captions and noise generated by EM and TAP compared to those in classification. In Fig. 2 (c), the learning decision space of UAP is much more complex, so its poor protection is to be expected.

### 3.3 Multi-step Error Minimization (MEM)

In the previous section, we showed that the model-based methods still fail to achieve effective protection due to the dispersion of image-text pairs. An intuitive strategy of enhancement entails optimizing both the image and the captions for a larger optimized space, boosting their convergence across different pairs in the feature space. Consequently, the optimized feature representations of the image and caption set exhibit similar distributions, facilitating the model's learning of their shortcut, as illustrated in Fig. 2 (d).

To this end, we take the EM method as the basic framework and propose adding an additional short text trigger to the caption to minimize contrastive loss, following the setting of an adversarial attack on text tasks [41]. Regarding the length of the trigger, longer triggers are more effective, while shorter triggers are more stealthy. Therefore, our method can be conceptualized as a tri-level iterative optimization problem, resembling a multi-step process of EM. Specifically, we sequentially optimize noise $\delta$ and text trigger $t$ to reduce contrastive loss between optimized images $I + \delta$ and optimized text $T \oplus t$, where $\oplus$ denotes the triggers that can be inserted into clean text $T$ at various positions. For simplicity, we choose to

add text triggers at the beginning of the text in this paper. As a result, our Multi-step Error Minimization (MEM) method can be formulated as follows:

$$\arg \min_{\delta \in \Delta, t \in \mathcal{T}} \mathbb{E}_{(I,T) \sim \mathcal{D}} \left[ \min_f \mathcal{L} \left( I + \delta, T \oplus t \right) \right]. \quad (4)$$

We can iteratively optimize the above problem sequentially by referring to the method in EM. We use projected gradient descent (PGD) [27] to solve the noise minimization problem in Eq. 4. Notably, to mitigate the noise overfit to the clean captions, we augment them by shuffling the clean caption in a batch and adding the correct matching text triggers on them. Thus, this generated noise can focus more on the text trigger than on part of the captions, when facing the semantic wrong captions. So we can get the optimal $\delta$ according to the following iterative formula,

$$\delta^{t+1} = \text{Proj} \left( \delta^t - \alpha \, \text{sign} \left( \nabla_{\delta^t} \mathcal{L} \left( I + \delta, \mathcal{S}(T) \oplus t \right) \right) \right), \quad (5)$$

where $\nabla_{\delta^t} \mathcal{L}$ is the gradient of the loss w.r.t $\delta^t$. $\alpha$ is the step size and $Proj()$ denotes the project of $\delta$ within the bound of the norm $(-\epsilon, \epsilon)$. $\mathcal{S}(\cdot)$ means shuffle the order of the clean caption samples in a batch with each iteration. For the text trigger minimization problem, we first initialize the trigger sequence by repeating the word "the" or "a" to the front of all inputs. In addition, we optimize the text trigger based on HotFlip [9], a method that approximates the effect of replacing a token using the gradient. We denote a text trigger as $t_i$ which is a hot vector and can be embedded in the form $e_i$. Thus, we can update the embedding for every trigger token $e_i$ to minimize the first-order Taylor approximation of CLIP loss around the current token embedding:

$$\arg \min_{\mathbf{e}'_j \in \mathcal{V}} \left[ \mathbf{e}'_j - \mathbf{e}_i \right]^\top \nabla_{\mathbf{e}_i} \mathcal{L}, \quad (6)$$

where $\mathcal{V}$ is the set of all token embeddings in the vocabulary and $\nabla_{\mathbf{e}_t}$ is the gradient of loss w.r.t $e_i$. We can calculate a set of candidate tokens $e'_j$ using a dot product with embedding of the vocabulary token and gradients. Finally, we can search for each optimal text trigger with a beam search in a set of candidate tokens. We consider the top-k candidates from Eq. 6 and search front to the last in every position in the trigger and score each beam using the loss on the current batch. We follow Wallace et al. [41] and use a small beam size for efficient computation. The pseudocode of our method is

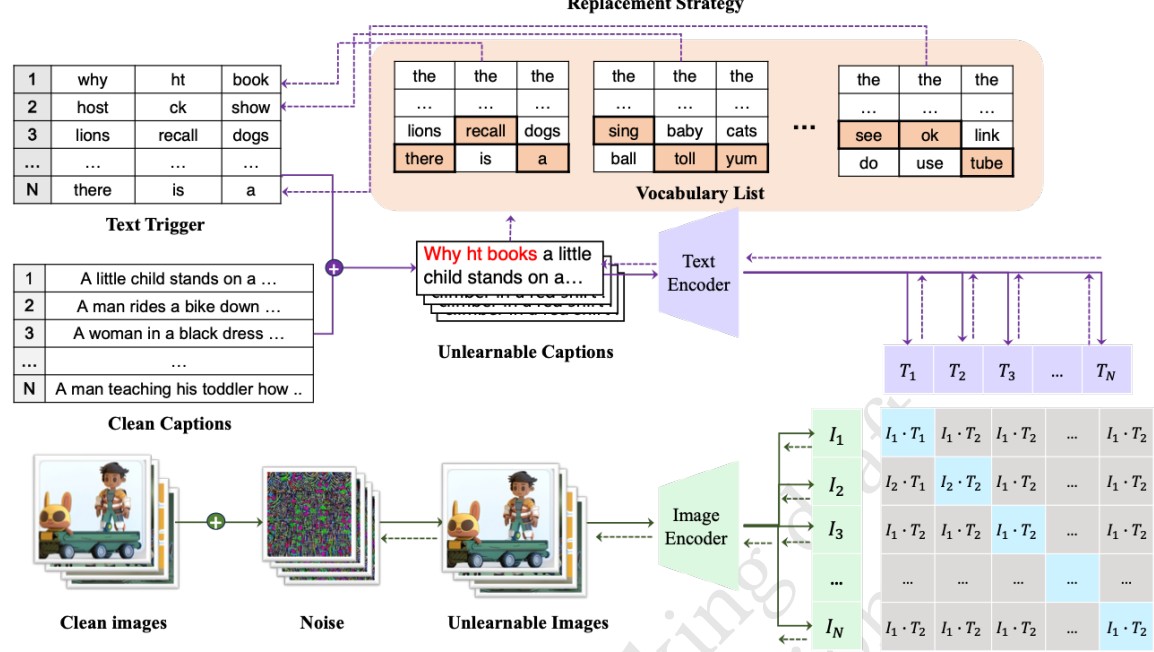

**Figure 3: The framework of MEM. At each step, we concatenate the current trigger to clean captions and attach noise to clean images. We then compute the gradient with current images and tokens. We update the images using Eq. 5 and update the text triggers with Eq. 6. The solid line represents forward propagation, and the dashed line represents backward propagation.**

shown in Algorithm 1. In Fig. 3, we can see the framework of the generation of multimodal unlearnable examples with our MEM.

## 4 EXPERIMENT

### 4.1 Experiment Setup

**Target models and datasets.** Following previous works [1, 42, 43], we adopt the open-source implementation of CLIP in the community[1]. For the architecture of the surrogate model, we choose the model with ResNet50 [19] as the image encoder and a Transformer [40] with certain architecture modifications serving as the text encoder. Our experiment involves three datasets: Flickr8K [44], Flickr30K [44], and MS-COCO [6]. Specifically, Flickr8K and Flickr30K consist of 8000 and 31,000 images, respectively, each accompanied by five captions. We adhere to the protocol established by [10, 22, 47], dividing the datasets into training/validation/testing sets with ratios of 6,000/1,000/1,000 and 29,000/1,000/1,000 for Flickr8K and Flickr30K, respectively. MS-COCO comprises 123,287 images, each annotated with five descriptions. As per [22], MS-COCO underwent division into 82,783 training images, 5,000 validation images, and 5,000 test images. Despite their smaller size compared to the large-scale datasets used to train the original CLIP model, these datasets remain suitable for storage and computational resources and have found widespread usage in MCL studies [42, 43].

**Training and Attack Setup.** Initially, we focus on training models from scratch in the main experiments and reserve discussion of the fine-tuning scenario to the case study on face privacy protection.

Following the training setup of the CLIP model in [1, 31], we choose a batch size of 128 and set the initial learning rate to 0.0005. The weight decay rate is set to 0.2, and we employ an Adam optimizer with decoupled weight decay regularization, while the learning rate decays using a cosine scheduler. We train these models from scratch on 1 A100 GPU for 32 epochs. During the attack stage, we extend the EM and UAP methods to MCL. In both cases, we assume that the surrogate models are CLIP models with the ResNet50, while the surrogate model for UAP is already trained on clean data. For our MEM, we choose an initial surrogate CLIP and iteratively optimize the noise and text trigger, stopping when the loss is below a threshold. We select a threshold of 0.01. Regarding the text trigger, we set the text sequence lengths as three and five, denoted MEM-3 and MEM-5, respectively. Typically, we set the noise bound of all methods with the $l_\infty$-norm $\epsilon = 8/255$.

**Evaluation Metrics.** We evaluate the impact of data protection by examining the decline in performance in image and text retrieval tasks, drawing from previous work on attacks in MCL [42, 43]. Two metrics are employed to illustrate the retrieval results. **Hit@10** measures the proportion of all target images/texts that appear within the top 10 of the list of rank. Higher Hit@10 values indicate that many text/image samples successfully retrieving target images/texts early, reflecting a better rank list. **Medr** refers to the median position of all target images / texts in the list of test images/texts. Lower Rank values signify earlier access to target images, indicative of a superior rank list. The performance of unlearnable multimodal examples is assessed using Hit@10 and Medr for image retrieval (Image → Text) and text retrieval (Text → Image) across all testing

**Table 1: Comparison of the effectiveness with different unlearnable examples on several datasets.**

| Dataset | Flick8k | | | | Flick30k | | | | MSCOCO | | | |
|---|---|---|---|---|---|---|---|---|---|---|---|---|
| Methods | Image → Text | | Text → Image | | Image → Text | | Text → Image | | Image → Text | | Text → Image | |
| | Hit@10 | Medr | Hit@10 | Medr | Hit@10 | Medr | Hit@10 | Medr | Hit@10 | Medr | Hit@10 | Medr |
| Random | 1.2 | 727 | 0.9 | 490 | 1.0 | 711 | 1.0 | 498 | 0.2 | 3419 | 0.2 | 2491 |
| Clean | 22.7 | 58 | 18.5 | 60 | 46.5 | 12 | 42.7 | 16 | 65.1 | 5 | 58.3 | 7 |
| EM [20] | 13.8 | 117 | 15.9 | 86 | 9.6 | 197 | 7.9 | 170 | 1.7 | 1213 | 1.6 | 880 |
| UAP [13] | 12.9 | 110 | 16.5 | 76 | 36.7 | 26 | 35.6 | 24 | 63.8 | 5 | 57.1 | 7 |
| MEM-3 (ours) | 4.1 | 304 | 3.7 | 250 | 3.8 | 412 | 2.2 | 308 | 1.7 | 1705 | 1.3 | 1301 |
| MEM-5 (ours) | **3.8** | **308** | **3.0** | **275** | **3.9** | **445** | **2.0** | **325** | **0.8** | **1883** | **1.1** | **1466** |

**Table 2: The transferability of MEM-3 generated on a ResNet50 model across different architectures models.**

| Dataset | Flick8k | | | | Flick30k | | | | MSCOCO | | | |
|---|---|---|---|---|---|---|---|---|---|---|---|---|
| Model | Image → Text | | Text → Image | | Image → Text | | Text → Image | | Image → Text | | Text → Image | |
| | $D_c$ | $D_u$ | $D_c$ | $D_u$ | $D_c$ | $D_u$ | $D_c$ | $D_u$ | $D_c$ | $D_u$ | $D_c$ | $D_u$ |
| RN50 | 58 | 308 | 60 | 275 | 12 | 445 | 16 | 325 | 5 | 1086 | 7 | 988 |
| RN101 | 52 | 236 | 58 | 197 | 10 | 404 | 17 | 306 | 5 | 989 | 6 | 844 |
| ViT-B/32 | 53 | 241 | 52 | 194 | 9 | 343 | 13 | 287 | 4 | 1064 | 4 | 920 |

images. Lower Hit@10 and higher Medr-Rank values signify more effective protection of the data.

## 4.2 Effectiveness and Transferability

In this section, we compare our method with extensions of existing popular unlearnable methods, including EM [20] and UAP [13]. Table 1 presents their retrieval results on different datasets. It is evident that UAP nearly fails to provide any protection for multi-modal data, while EM demonstrates a certain level of protection. Moreover, as the size of the dataset increases, the effect of EM improves due to the denser caption distribution in the feature space, consistent with our analysis in Section 3.2. However, our MEM consistently offers robust protection for multimodal data, reducing retrieval performance to nearly half random guessing. In particular, MEM-5, with a longer text trigger, obtained greater results in reducing the performance of the hacker model compared to MEM-3, likely because longer text triggers enable the model to focus more effectively on the trigger and establish a shortcut.

Fig. 4 illustrates the descending curves of the training loss trained on unlearnable examples generated by different methods and the retrieval Medr on the clean test set. From (a), we observe that although EM enables the loss to fall faster compared to normal training, our method, MEM-3 and MEM-5, have a smaller loss on the first epoch, suggesting that the model can quickly learn the shortcuts. The UAP hardly speeds up the loss decline significantly, so it is understandably inefficient. From (b), we find that all models are trained with Medr decreasing compared to random guessing, but the model trained on unlearnable examples generated by MEM stops learning the fastest, reaches the worst retrieval result, and does not learn better further as the epoch increases. The above observations are consistent with the results in Table 1.

We assume that data protection is a completely black-box setting, wherein the protector lacks knowledge of the hacker model's architecture. Thus, we evaluate the performance of our MEM generated on the ResNet50 surrogate model on different hacker models,

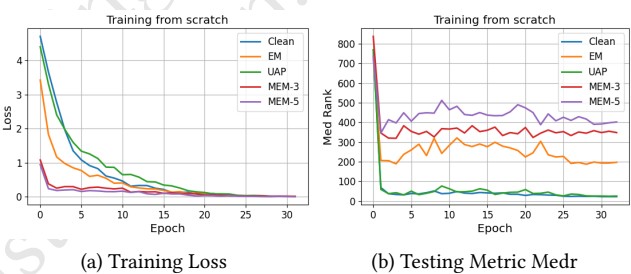

(a) Training Loss      (b) Testing Metric Medr

**Figure 4: Training loss curves and Testing metric Medr curves on Flick30K with different methods.**

including ResNet101, and ViT-B/32. The results are presented in Table 2. We find that these examples can be successfully transferred across different models and can degrade the performance of the CLIP model. Although they generated a surrogate model with a structure different from that of the hacker model can also exhibit the same protective effect, this observation aligns with the one of unlearnable examples in image classification. This is likely because the feature preferences learned by the models are similar.

## 4.3 Analysis

**Attention of the models.** Fig. 5 presents the heatmaps of images and text, illustrating the attention of models trained on clean and unlearnable examples. The Grad-CAM [35] is utilized to visualize the model attention for images, while the Integrated Gradients [38] is employed to visualize the attention to the text. Lighter colors represent higher attention from the model. Remarkably, for the image, models in (a), (b), and (c) all focus on the central region, which correlates with the captions. In contrast, model (d), trained on samples generated by MEM-3, fails to accurately recognize the clean image due to only learning noise features. Similarly in the text, models in (a), (b), and (c) all focus on the keywords of 'glass', while the model in (d) put the attention on the first three words, probably

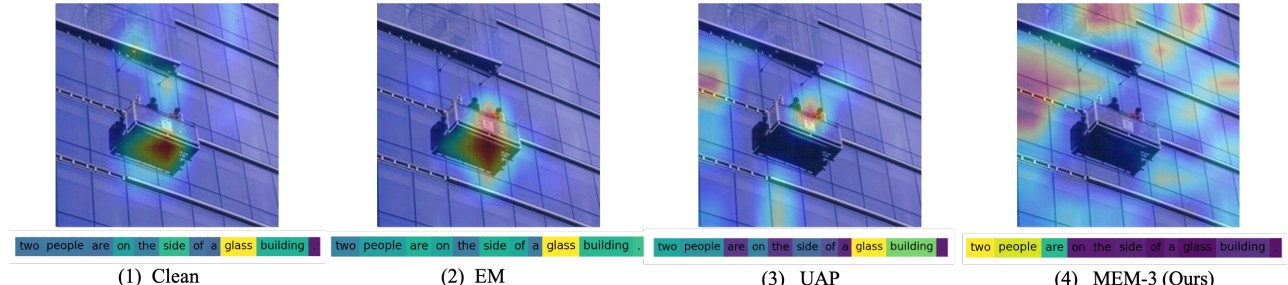

(1) Clean                     (2) EM                     (3) UAP                     (4) MEM-3 (Ours)

**Figure 5: Attention Maps Visualization: comparing four models on clean data and unlearnable examples with different methods.**

**Table 3: The protection effect of including different percentages of unlearnable examples datasets on Flick8K.**

| Percentage Poisoning ($p$) | 100% Poisoning | | 90% Poisoning | | 50% Poisoning | | 20% Poisoning | |
|---|---|---|---|---|---|---|---|---|
| | Image → Text | Text → Image | Image → Text | Text → Image | Image → Text | Text → Image | Image → Text | Text → Image |
| MEM-3 | 304 | 250 | 110 | 95 | 96 | 98 | 91 | 97 |
| MEM-5 | **308** | **275** | 163 | 126 | 141 | 114 | 135 | 116 |

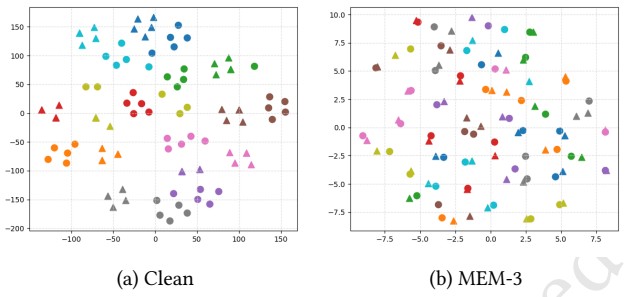

(a) Clean                     (b) MEM-3

**Figure 6: t-SNE visualization of the clean samples and unlearnable examples for clean model and poisoned model.**

because MEM-3 always optimizes the noise and the first three text triggers to create shortcuts. These visualizations report that the EM and UAP are ineffective enough in protecting the multimodal data, while MEM-3 has an obvious effectiveness.

**Visulaization of unlearnable examples.** We visualize the feature distributions of clean samples under the normal model and the feature distributions of unlearnable examples optimized by MEM-3 on the unlearned model in Fig 6. We represent image features with triangles and text features with circles, with the same color indicating five identical but transformed images and their corresponding different descriptions in the dataset. From (a), we observe that under the clean model, the same images and texts cluster together internally, and the corresponding image-text pairs are close to each other. However, in (b), there is a divergence between the same images and texts, with only the pairs being pairwise close to each other. This indicates that our methods effectively facilitate the model in learning the shortcuts between noise and textual triggers.

**The fraction of unlearnable data.** In the real world, hackers may not be able to obtain data from only one source, so the dataset may not contain only unlearnable examples. Here, we test the effect of unlearnable examples at different proportions on Flickr8K with MEM-3. Table 4 shows, our protection power gradually diminishes as the percentage of unlearnable examples decreases. As identified by previous work in classification [13, 20], the effect of them will

quickly drop when the data are not 100% poisoned, and as the number decreases to 50% they have almost no effect. In the case of MCLs, when they only constitute 20% of the dataset, some level of protection is still observed. This difference may be attributed to the dispersion of image-text pairs, unlike in classification where clean images may correct the images with perturbation, resulting in a rapid decline in protection efficacy. This finding underscores the superior robustness of unlearnable multimodal examples.

### 4.4 Extension with Semantic Triggers

In the previous methodology, we propose to use the gradient by the Hotflip method to select embeddings in the vocabulary list with large inner products to serve as candidate words for replacement. However, this approach may result in the generation of candidate words lacking semantic information, potentially impacting social reading or being detected and removed by hackers. Therefore, here we leverage the pre-training parameters of the BERT model to generate some semantically relevant substitutions as vocabulary lists. Subsequently, we embed these substitutions to obtain $\mathcal{V}_{bert}$, and then compute the inner product with the gradient and token embeddings of the substitutions, which yields replaced words with semantics. Table 4 displays examples of text triggers generated by our MEM-3 method under different datasets, including those generated with and without the BERT method. Text triggers generated with BERT exhibit greater naturalness and semantic coherence compared to those generated without BERT. Additionally, we show the Medr results of these triggers in text-image retrieval. It is observed that triggers generated without BERT have better protection performance, which is attributed to their larger search space, while triggers generated with BERT are restricted to a vocabulary space $\mathcal{V}_{bert}$, resulting in inferior protection efficacy.

### 4.5 Case Study: Face-Privacy Protection

Here, we conduct a case study to apply our MEM noise to a real-world scenario: protecting personal face images and associated information on social media platforms, such as names. This scenario is arguably one of the most common multimodal privacy

**Table 4: Comparison of text triggers generated by MEM-3 method with and w/o BERT, and their effect on protection.**

| Dataset | | Captions (**red** = text trigger) | Image → Text | Text → Image |
|---|---|---|---|---|
| Flickr8k | w/o Bert | **And lopez ...** a girl in a pink shirt has jumped into the air. | 304 | 250 |
| | with BERT | **Next, suddenly** a girl in a pink shirt has jumped into the air. | 256 | 213 |
| Flickr30k | w/o BERT | **The baked instaa** bride having a photo shoot outdoors around lots of people. | 412 | 308 |
| | with BERT | **I was like** a bride having a photo shoot outdoors around lots of people. | 365 | 278 |
| MSCOCO | w/o BERT | **Conde matsu grinding** a man and two dogs in the snow. | 1705 | 1301 |
| | with BERT | **There he saw** a man and two dogs in the snow. | 1356 | 1028 |

**Table 5: The protection effect of unlearnable examples generated on ResNet50 fine-tuning on different pre-trained models.**

| Model | RN50 | | | | RN101 | | | | ViT-B/32 | | | |
|---|---|---|---|---|---|---|---|---|---|---|---|---|
| Metric | Image → Text | | Text → Image | | Image → Text | | Text → Image | | Image → Text | | Text → Image | |
| | Hit@10 | Medr | Hit@10 | Medr | Hit@10 | Medr | Hit@10 | Medr | Hit@10 | Medr | Hit@10 | Medr |
| Pre-trained | 2.6 | 77 | 2.6 | 77 | 3.3 | 83 | 3.3 | 83 | 3 | 87 | 3 | 87 |
| Fine-tuned | 80.7 | 1 | 80.7 | 1 | 85.3 | 1 | 85.3 | 1 | 94 | 1 | 94 | 1 |
| MEM-3 | 6.7 | 74 | 6.7 | 74 | 6 | 66 | 6 | 66 | 93.3 | 1 | 93.3 | 1 |
| MEM-5 | 8.7 | 82 | 8.7 | 82 | 12.7 | 58 | 12.7 | 58 | 91.3 | 1 | 91.3 | 1 |

protection scenarios. In previous sections, we assumed that hackers train models with multi-modal unlearnable examples from scratch. However, in most real-world scenarios, hackers will opt to fine-tune pre-trained models provided by the community. Therefore, here we assume that the protector aims to prevent their facial images and names from being fine-tuned with a pre-trained CLIP model. We adopt the Open AI's released parameters to initialize our pre-trained models. We still assume the protector has access to their image and text data to generate multimodal unlearnable examples before sharing them on on-line social media platforms. However, they are unable to know the hacker's model architecture and the parameters of pre-trained models. In this black-box scenario, we explore whether our method can still successfully prevent the CLIP model from learning the protector's face images.

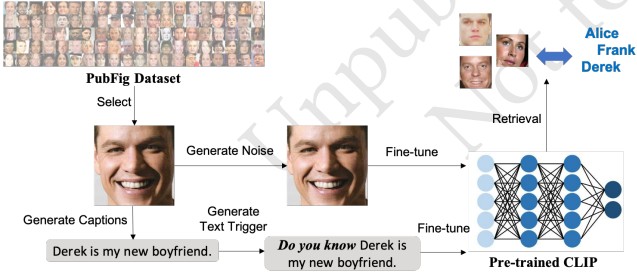

**Figure 7: Illustration of the face-privacy protection pipeline.**

We conducted experiments using the PubFig database [23], a large real-world face dataset comprising 58,797 images of 200 individuals collected from the internet. For retrieval evaluation, we randomly selected one photo of each celebrity as the test set and utilized all remaining images for training. It's noteworthy that the pre-trained CLIP model already possesses knowledge of the real names and faces of these celebrities. Therefore, for authentic fine-tuning, we altered their names and provided a set of text templates relevant to the name for caption generation, as shown in Fig. 7. Subsequently, we generated unlearnable examples using MEM and

employed a pre-trained surrogate model with ResNet50 as the image encoder. We fine-tune the pre-trained models for 10 epochs and a small learning rate of 0.00001 and evaluate them with different hacker models. The results are presented in Table 5. For comparison, we also test the results of the pre-trained models and the benign fine-tuned models. Table 5 shows that the initial model shows poor performance on the test set, since the face and name features are not aligned. However, with only 10 epochs of fine-tuning, the Medr of its retrieved results reaches 1. This may be because of the strong feature representation of the pre-trained model, enabling it to swiftly adapt to the new task during the fine-tuning process. However, our MEM approach can prevent these fine-tuned models from learning the correlation between face and name features, thereby impeding accurate person retrieval on the test set. Our unlearnable examples were generated using the ResNet50 surrogate model and effectively protect against attacks on the ResNet101 hacker model. However, their efficacy is considerably reduced on Vision Transformer, possibly due to significant variations in architecture and initialization parameters between models.

## 5 CONCLUSION

In this paper, we explore multimodal data protection, particularly focusing on image-text pairs, wherein we generate multimodal unlearnable examples to prevent exploitation by MCL. We extend previous classification methods to this context, revealing their limitation in MCL due to the increased modality and data dispersion. In light of these findings, we introduce a novel generation approach named Multi-step Error Minimization (MEM), which is based on the framework of EM. MEM effectively establishes a shortcut between noise and textual triggers and demonstrates transferability across different hacker models. Additionally, we utilize various visualization tools to validate the effectiveness of our approach. Finally, we present a case study on face-privacy protection aimed at preventing fine-tuning on pre-trained models with personal faces and identifying information. Our work opens up a new direction, and our methods are expected to be applicable to other modal pairs, such as audio-text and audio-image pairs.

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
