# OpenReview forum: "Multimodal Unlearnable Examples: Protecting Data against Multimodal Contrastive Learning"
_acmmm.org/ACMMM/2024/Conference — MM2024 Poster_

### Official Review · Reviewer_2b8t · 2024-04-27

**Rating:** 2
**Confidence:** 4

**Summary:**

This paper presents an unlearnable example method for protecting the privacy of image-text pairs from being learned by DNNs in multi-modal scenarios. Experimental results on benchmark datasets demonstrate the effectiveness of the proposed approach.

**Strengths:**

1.The novelty of the scenario. The consideration of unlearnable examples in multi-modal contexts in this paper is quite novel to me, and it holds significant practical implications in real-world scenarios.

2.This paper is well-written and easy to understand.

**Limitations:**

1.The puzzling Fig 1. The authors claim in Fig 1 that the proposed unlearnable example can prevent the model from learning features of private information. But what if it's proven that the model hasn't learned these features of private information? How can we be certain of this? There are also no relevant metrics in the experiments to assess this.

2.Inaccurate statement. Lines 122-123 and Lines 251-252 incorrectly equate the concepts of availability/indiscriminate poisoning attack with the unlearnable example attack. In fact, availability/indiscriminate poisoning attack should be categorized based on the attack's objective, where any attack resulting in a decrease in final test accuracy is termed as availability/indiscriminate poisoning attacks, including clean-label scenarios (equivalent to unlearnable examples) and dirty-label scenarios [1].

3.Self contradictory statement. In Lines 157-158, the authors claim that the methods for unlearnable examples in image classification all fail to be effective in multimodal data, but it is evident from Table 1 that the transferred methods EM and UAP are effective.

4.Time complexity of proposed approach. In Lines 167-168, the authors mention the importance of more efficient generation of shortcut for multimodal data. Therefore, I am curious about the efficiency of the proposed solution and its comparison with other methods. However, I did not find any discussion on efficiency in the experimental section.

5.Confused definition. In Lines 226-228, the authors define the target of data poisoning attacks as the decrease in accuracy for specific samples, which is not the objective of indiscriminate/availability poisoning attacks. The indiscriminate/availability poisoning attacks aim to indiscriminately reduce the accuracy of samples in the test set.

[1] Support vector machines under adversarial label noise. In: Proceedings of the 3rd Asian Conference on Machine Learning. (ACML’11)

**Suitability:**

3

---

### Official Review · Reviewer_xgcj · 2024-05-04

**Rating:** 5
**Confidence:** 2

**Summary:**

This paper proposes MEM (Multi-step Error Minimization), a new method to create unlearnable examples for privacy protection in Multimodal Contrastive Learning. MEM builds on the EM framework to optimize image noise and text triggers, hindering models from learning the shortcut between the noise and the trigger.The article is well-written, with clear and concise language that is easy to follow.

**Strengths:**

1. I like Figure 1, as it provides an excellent and clear explanation.
2. The article is well-written, with clear and concise language that is easy to follow.
3. The approach of simultaneously optimizing images and text to generate unlearnable examples is novel and demonstrates effective utilization of knowledge in the multimedia domain.

**Limitations:**

1. I suggest moving the pipeline up and spending more space introducing it. In the current version, it is challenging for readers to grasp the implementation details of the proposed method. The methodology section should be expanded to provide more elaborate explanations.
2. In line 494, there is no Algorithm 1.
3. Given the existing exploration of defending against availability attacks in classification tasks, I believe it would be beneficial to the community to include a discussion or supplement on how to defend against availability attacks in multimodal tasks.

**Suitability:**

2

---

### Official Review · Reviewer_7ktv · 2024-05-23

**Rating:** 4
**Confidence:** 4

**Summary:**

In this paper, the authors propose a novel approach to protect multimodal data, specifically image-text pairs, from unauthorized use in Multimodal Contrastive Learning (MCL) models. The authors propose a Multi-step Error Minimization (MEM) method that generates unlearnable examples to prevent models from learning private information. The effectiveness of the MEM approach is demonstrated through extensive experiments.

**Strengths:**

1.	The paper introduces a new scenario for multimodal data protection against MCL, addressing a gap in current research.
2.	The paper provides a detailed analysis of the limitations of previous methods when applied to multimodal data, offering clear insights into the challenges and their proposed solutions.
3.	This paper is easy to follow.

**Limitations:**

1.	The paper does not discuss the efficiency of the proposed MEM method compared to existing methods, particularly in terms of computational complexity and resource requirements.
2.	The approach focuses on image-text pairs, which may limit its applicability to other types of multimodal data such as audio-visual pairs or text-sensor data combinations.

**Suitability:**

2

---

### Meta-Review · Area_Chair_c71y · 2024-07-02

**Recommendation:** Accept (Poster)
**Confidence:** 4

**Metareview:**

A good problem to study and the approach was deemed novel by reviewers.
Two out of three reviewers voted for acceptance, while the other one has reservations. However, looking at that reviewer's remaining concerns, they do not seem to be fundamental, technical issues. Hence I'm siding with the majority opinion.